# Genetic and epigenetic variation in the lineage specification of regulatory T cells

**Aaron Arvey[1]\*, Joris van der Veeken[1], George Plitas[2,3], Stephen S Rich[4], Patrick Concannon[5], Alexander Y Rudensky[1,2]\***

[1]Immunology Program, Howard Hughes Medical Institute, Memorial Sloan Kettering Cancer Center, New York, United States; [2]Ludwig Center for Cancer Immunotherapy, Memorial Sloan Kettering Cancer Center, New York, United States; [3]Breast Service, Memorial Sloan Kettering Cancer Center, New York, United States; [4]Center for Public Health Genomics, Department of Public Health Sciences, Division of Biostatistics and Epidemiology, University of Virginia, Charlottesville, United States; [5]Genetics Institute, Department of Pathology, Immunology and Laboratory Medicine, University of Florida, Florida, United States

**Abstract** Regulatory T (Treg) cells, which suppress autoimmunity and other inflammatory states, are characterized by a distinct set of genetic elements controlling their gene expression. However, the extent of genetic and associated epigenetic variation in the Treg cell lineage and its possible relation to disease states in humans remain unknown. We explored evolutionary conservation of regulatory elements and natural human inter-individual epigenetic variation in Treg cells to identify the core transcriptional control program of lineage specification. Analysis of single nucleotide polymorphisms in core lineage-specific enhancers revealed disease associations, which were further corroborated by high-resolution genotyping to fine map causal polymorphisms in lineage-specific enhancers. Our findings suggest that a small set of regulatory elements specify the Treg lineage and that genetic variation in Treg cell-specific enhancers may alter Treg cell function contributing to polygenic disease.

\*For correspondence: aarvey@cbio.mskcc.org (AA); rudenska@mskcc.org (AYR)

**Competing interests:** The authors declare that no competing interests exist.

## Introduction

Lineage specification factors promote cellular differentiation by binding DNA regulatory elements to stably alter the local chromatin state and affect gene transcriptional outputs (*Visel et al., 2009*; *Thurman et al., 2012*; *Nord et al., 2013*; *Ostuni et al., 2013*). Since the majority of protein-coding genome sequence is under constraint, due to the cost of deleterious mutational perturbations in protein function, changes in non-coding regulatory portions of the genome are thought to serve as a major source of evolutionary innovation and substrates for selection (*King and Wilson, 1975*; *Schmidt et al., 2010*; *Goncalves et al., 2012*; *Ward and Kellis, 2012*). Histone modifications and chromatin accessibility, which serve as proxies for epigenetic transcriptional control, can exhibit conservation shared with the underlying genetic elements in a given cell lineage across species and within the genetically diverse human population (*Kasowski et al., 2013a*; *Kilpinen et al., 2013a*; *Long et al., 2013*; *McVicker et al., 2013a*; *Stergachis et al., 2014*; *Vierstra et al., 2014*). However, it remains unknown if changes in the chromatin state of regulatory elements associated with cell lineage specification are conserved in their lineage-specificity (*Odom et al., 2007*). That is, it is unknown to what extent regulatory elements are cell lineage-specific across multiple organisms and individuals. Strong conservation would imply that the differentiation process of a specialized cell population is highly constrained at a genomic level, whereas extensive variation would suggest that lineage specification is dependent on only a few immutable genetic regulatory elements.

**eLife digest** The immune system protects the body from infection. Key to this protection is the ability to mount an immune response that is sufficient to deal with the threat, but is not so large that the damage it causes to the body exceeds its immediate benefit. Immune cells called regulatory T cells (or Treg cells for short) help to shut down the immune response after a threat has been successfully destroyed. They also prevent the immune system from attacking the body's own cells, a phenomenon known as autoimmunity.

All cells in the body carry the same set of genes, but the activity of these genes varies between cell types to enable the cells to perform their different jobs. This is possible because our DNA contains regions called regulatory elements that can control the expression of particular genes. These regions can be activated in specific types of cells, which results in specific chemical modifications to DNA that only affect gene activity in those cells.

The DNA sequences of these regulatory elements vary between individuals. This 'genetic variation' can lead to differences in the chemical modifications that occur to DNA, which is known as epigenetic variation. This means that Treg cells in one person may work in a different way to those in another individual, which could make some individuals more susceptible to autoimmune diseases than others. However, it was not clear how much genetic and epigenetic variation exists in Treg cells.

Here, Arvey et al. examined Treg and other immune cells from human and mouse donors. The experiments show that some of the epigenetic modifications present in many individuals only in Treg cells, which indicates that they may be important for the activity of the Treg cells. Unexpectedly, most of the epigenetic modifications were specific to either mice or humans, but Arvey et al. identified a core set of genes that had been modified only in Treg cells in both species. In the human cells, Arvey et al. also identified genetic differences in regulatory elements that are associated with autoimmune diseases.

Arvey et al.'s findings suggest that a small set of regulatory elements are crucial to the role of Treg cells in the immune system. Furthermore, genetic variation in these elements can lead to epigenetic changes in Treg cells that contribute to autoimmune diseases in humans. Further study may lead to the development of new treatments for these diseases.

We examined conservation of chromatin states and gene expression in human and mouse regulatory T (Treg) cells, a key cell population required for maintenance of tolerance to 'self' and restrained inflammatory responses during infection. Treg cell differentiation and function is controlled by a late-acting lineage specification factor Foxp3, which is induced upon signaling through T cell (TCR) and cytokine receptors (*Hill et al., 2007*; *Ohkura et al., 2012*; *Samstein et al., 2012*). Treg cells generated in the thymus exhibit a 'naive' phenotype and display low, if any suppressor activity in contrast to activated Treg cells, which acquire highly potent suppressor function as the result of TCR signaling-dependent activation and division (*Battaglia and Roncarolo, 2009*; *Levine et al., 2014*; *Vahl et al., 2014*). Genetic Foxp3 deficiency or elimination of Treg cells in mice results in lymphoproliferation and myeloproliferation leading to widespread fatal autoimmune lesions largely driven by activation of 'self'-, commensal microbiota-, and food-derived antigen-reactive CD4 T cells (*Brunkow et al., 2001*; *Fontenot et al., 2003*; *Khattri et al., 2003*; *Kim et al., 2007*). Human IPEX patients with loss-of-function Foxp3 mutations and congenital Treg cell deficiency present with neonatal diabetes, thyroiditis, autoimmune anemia and neutropenia, autoimmune hepatitis, exudative dermatitis, enteropathies, and hyper-IgE syndrome (*Bennett et al., 2001*; *Torgerson and Ochs, 2007*; *Verbsky and Chatila, 2013*).

To characterize evolutionary conservation and human-to-human genetic variation that underlies the lineage specification of Treg cells, we investigated the active regulatory DNA elements and gene expression in murine and human resting and activated Treg cells and compared them to their counterpart resting naive CD4$^+$ T cells (Tn) and activated and effector CD4$^+$ T cells (Teff). The analysis of orthologous lineage-specific epigenetic features revealed that while the vast majority of regulatory loci were genetically conserved, the lineage-specific epigenome was markedly different in mouse and human, supporting a model whereby few conserved elements contribute to the specification of the Treg cell lineage. A subset of the conserved lineage-specific elements contained nucleotide

polymorphisms associated with altered epigenetic activity in human Treg cells isolated from a small cohort of healthy donors. Finally, we analyzed the polymorphic conserved elements using high-resolution epigenetic and genome-wide genetic disease association data and were able to localize disease-risk linked polymorphisms to regulatory loci that are exclusively epigenetically active in Treg cells. Our results support a role for Treg dysfunction in common polygenic diseases.

## Results

### Genetic and epigenetic conservation of regulatory non-coding DNA elements between human and mouse CD4$^+$ T-cell lineages

To characterize the epigenetic state of human Treg and CD4$^+$ T-cell populations, we isolated CD4$^+$ T cells from the peripheral blood of individual anonymous human donors using negative selection (see 'Materials and methods'). Previously characterized subsets of Treg cells and naive and effector CD4$^+$ T cells were purified using FACS sorting based on expression of CD25 and CD45RO (>97–99% purity) (*Figure 1A–C*, *Figure 1—figure supplement 1A,B*). Mouse resting Treg cells and CD4$^+$ T cells were isolated from phosphate-buffered saline (PBS)-treated *Foxp3$^{DTR-GFP}$* mice using FACS sorting based on the expression of GFP-DTR (diphtheria toxin (DT) receptor) fusion protein or lack thereof, respectively. In these mice, DNA sequence encoding IRES-driven GFP-DTR fusion protein was inserted in frame into the 3′ UTR of the endogenous *Foxp3* gene. These knock-in mice enabled isolation of Treg cells based on GFP expression and their depletion upon DT injection (*Kim et al., 2007*). The corresponding populations of activated CD4$^+$ Teff and Treg cells were isolated from *Foxp3$^{DTR-GFP}$* mice subjected to transient ablation of Treg cells followed by their recovery and activation in response to inflammation on day 11 after administration of a single dose of DT as described (see 'Materials and methods'). We desired to compare aTreg vs Teff in addition to Treg vs Tn cell populations, since they have comparable antigen experience; however, human and mouse T-cell subsets isolated ex vivo may have experienced different in vivo activation conditions. Therefore, we compared activated Treg lineage-specific transcriptional and epigenetic features to those of conventional T effector populations for each organism to account for the species-specific activation associated changes. In total, we analyzed 16 human cell samples (7 donors: 7 aTreg, 4 rTreg, 2 Teff, 2 Tmem, and 1 Tn samples) and 10 murine samples (2 aTreg, 4 rTreg, 2 Teff, and 4 Tn biological replicates independently isolated from different mice).

To identify active regulatory elements of the CD4$^+$ T-cell epigenome, we performed chromatin immunoprecipitation (ChIP) of histone H3 acetylated at lysine 27 (H3K27ac) followed by high-throughput sequencing (ChIP-seq). This histone modification serves as a reliable marker for active regulatory elements (*Creyghton et al., 2010*; *Arvey et al., 2012*). We observed ~31,000 H3K27 acetylation peaks across the genome in CD4$^+$ T-cell subsets, which we stratified by reads aligned per million (RPM) in each cell population. In addition to histone acetylation, we incorporated previously generated DNase-seq hypersensitive site (DHS) data sets to enable higher resolution positioning of active acetylated regulatory elements (*Arvey et al., 2012*; *Samstein et al., 2012*; *Thurman et al., 2012*; *Epigenome Roadmap Consortium, 2015*).

We found that the overall epigenetic features of chromatin in human- and mouse-activated Treg cells were highly conserved based on qualitative and quantitative analyses. Human and murine loci with sufficient read counts of H3K27ac were 'lifted over' and merged to form a single set of peaks that exist in either organism (see 'Materials and methods' and *Supplementary files 1-6* for details). This analysis captures micro- and macro-genetic differences, including sequence homology, insertions/deletions, inversions, and chromosome breaks (*Figure 1—figure supplement 1C–F*). We identified loci that were genetically and epigenetically conserved (e.g., *LRRC32* [encoding GARP]; *Figure 1D*), epigenetically active only in a single organism due to unique genetic elements (e.g., *YY1*; *Figure 1E*), and with loss or gain of histone modifications at genetically conserved sites (e.g., *LRRC32* upstream gene *C11ORF30*, *Figure 1D*, *Figure 1—figure supplement 1G*). Genome-wide comparison between human and mouse identified ~27,000 orthologous regulatory elements (*Figure 1F,G*, *Figure 1—figure supplement 1H,I*). The level of H3K27ac at conserved genetic elements was highly correlated ($r = 0.48$), with a plurality of shared elements being weakly active in both organisms (*Figure 1G*). A similar correlation was revealed by analyses of chromatin accessibility at DHSs across organisms (*Figure 1—figure supplement 1J*).

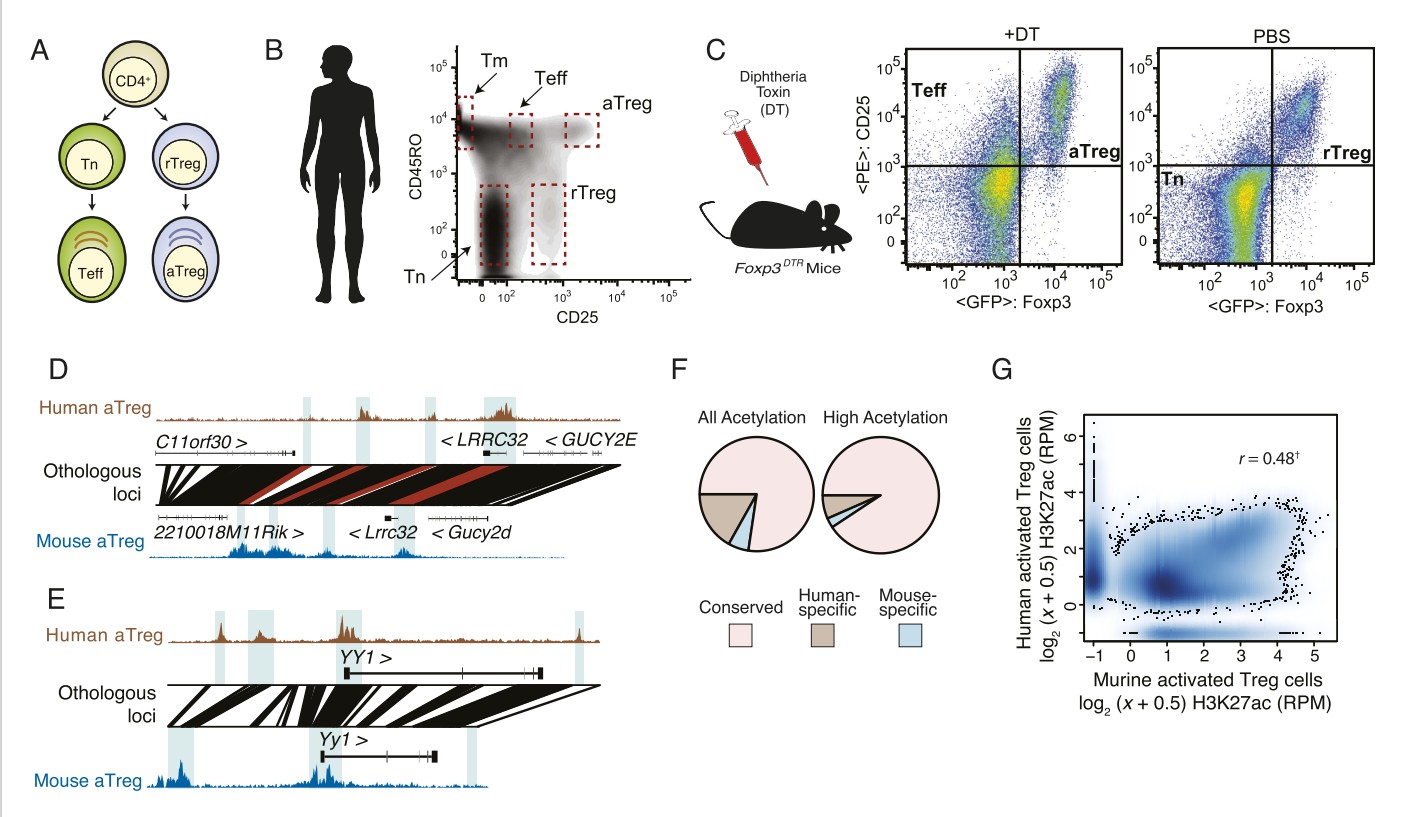

**Figure 1**. Analysis of genetic and epigenetic conservation in mouse and human Treg and CD4+ T cell subsets. (**A**) Schematic representation of profiled CD4+ T-cell subsets. Abbreviations: naive T cell (Tn); effector T cell (Teff); resting regulatory T cell (rTreg); activated regulatory T cells (aTreg). (**B**) The indicated human CD4+ T-cell subpopulations were FACS sorted based on CD3, CD4, CD45RO, and CD25 expression from preparations of peripheral blood mononuclear cells (PBMCs) from healthy human donors. Highly purified Treg cell subpopulations were obtained using a FACS Aria II fluorescent cell sorter (*Figure 1—figure supplement 1A*). Epigenetic profiling was performed using the following 16 cell samples isolated from 7 healthy donors: including 7 aTreg, 4 rTreg, 2 Teff, 2 Tmem, and 1 Tn independently isolated cell populations. See also *Figure 1—figure supplement 1A,B*. (**C**) Resting and activated murine CD4+ T-cell subpopulations were FACS sorted from *Foxp3*[DTR-GFP] mice injected with PBS or diphtheria toxin (DT), respectively. In *Foxp3*[DTR-GFP] mice, Treg cells express diphtheria toxin receptor (DTR). Mice injected with DT underwent punctual Treg cell depletion and consequent transient systemic inflammation, which resulted in activation of rebounding Treg and conventional T cells. A total of 10 mouse cell samples isolated using FACS sorting from DT-treated and DT-untreated *Foxp3*[DTR] mice were analyzed: 2 aTreg, 4 rTreg, 2 Teff, and 4 Tn biological replicates. (**D**, **E**) Genetic and epigenetic conservation at select loci. Multiple regulatory elements near *LRRC32* and *YY1* are genetically and epigenetically conserved: *YY1* has two epigenetic elements that are not conserved in human; *LRRC32* has a regulatory element that is genetically, but not epigenetically conserved. (**D**) Acetylation at the *LRRC32* locus shows multiple conserved genetic elements that illustrate concordant and discordant epigenetic states across species (highlighted regions). The human *LRRC32* locus (top) and murine *Lrrc32* locus (bottom) feature extensive genetically orthologous elements (lines connecting human and murine genomic coordinates) containing species-specific insertions/deletions (white space). H3K27ac ChIP-seq reads per million (RPM) are shown on y-axis for the indicated species and cell lineages. Orthologous regions with regulatory elements of interest are shown by blue background highlighting and red connecting lines. A genetically conserved element near *LRRC32* is epigenetically active in mouse, but not in human (leftmost highlighted region). (**E**) Two regulatory elements near *YY1* are epigenetically active in human but are not genetically conserved in mouse (leftmost and rightmost highlighted regions). (**F**) Genome-wide fractions of genetically conserved acetylated loci. Loci with high read counts are more frequently genetically conserved (shown) as are regulatory elements more proximal to gene body (*Figure 1—figure supplement 1H*). (**G**) Genome-wide quantification of epigenetic conservation. Axes show H3K27ac quantification (reads per million: RPM) of murine (x-axis) and human (y-axis) acetylated loci. Qualitatively, the vast majority of regulatory elements are epigenetically conserved in mouse and human Treg cells, with genome-wide quantitative correlation of r = 0.48 ('†' indicates that correlation is computed only for genetically conserved loci; non-conserved loci are shown on axes and by definition cannot be epigenetically conserved). Correlation across mouse biological replicates was r > 0.99 and between human donors r > 0.94, indicating that the observed conservation and lack thereof are reflective of biology and not technical/replicate reproducibility (*Figure 1—figure supplement 1K*).

The following figure supplement is available for figure 1:

**Figure supplement 1**. Quality analysis of epigenetic datasets.

## Genetic and epigenetic conservation of the Treg cell lineage specification program

We reasoned that the most critical genetic components of the Treg lineage identity would be genetically and epigenetically conserved in mouse and human and additionally be Treg lineage-specific in both species. We thus characterized regulatory elements with conserved lineage-specific activity, which we defined as increased or decreased H3K27ac amounts in Treg cells in comparison to non-Treg cells in both mouse and human. This was an extension of the above analyses in which we characterized genetic conservation of regulatory elements and the epigenetic activity of these elements in mouse and human Treg cells. Previous studies of multiple cell lines and specialized tissues from different organisms have shown that non-coding regulatory elements are more likely to be genetically conserved and active based on their epigenetic features (*Bernstein et al., 2005*; *Schmidt et al., 2010*; *Woo and Li, 2012*). However, it remained unclear whether cell lineage-specific regulatory elements, which define identity and function of a given cell type, are conserved at genetic, epigenetic, and lineage-specific functional levels (*Cheng et al., 2014*; *Vierstra et al., 2014*; *Yue et al., 2014*).

We identified a handful of regulatory elements with cell lineage-specific acetylation in both human and mouse. For instance, the *LRRC32* locus, which encodes the GARP protein that acts as a functionally important marker of activated Treg cells in humans (*Wang et al., 2009*), has multiple regulatory elements that are genetically conserved, epigenetically conserved, and only active in the Treg cell lineage (*Figure 2A*). Intriguingly, while the genome-wide epigenetically active landscape was similar in human and mouse, quantification of cell lineage specificity revealed that very few of the elements are Treg cell-specific in both species (*Figure 2B* and *Figure 2—figure supplement 1A–C*). Specifically, we were able to identify conserved epigenetic modifications at a small set of regulatory elements, which could be assigned to a handful of genes, including *FOXP3*, *IL2RA* (*CD25*), *CTLA4*, *IKZF2* (*HELIOS*), *IL2RB*, *DUSP4*, *BLIMP1*, *TNFRSF8*, *TNFRSF9*, *CCR8*, and *LRRC32* (*Figure 2—figure supplement 1D,E*). Interestingly, there was greater statistical enrichment for conserved loci downregulated in a Treg-specific fashion, which included *IL7R* (*CD127*), *IL2*, *PDE3B*, *LEF1*, *IL21*, *PDE7A*, *TNFSF8*, and *THEMIS*.

To assess potential functional relevance of species-specific features of H3K27 acetylation, we analyzed genome-wide RNA expression levels in the human and mouse Treg and CD4[+] T-cell populations (*Figure 2C*; *Supplementary files 10-12*). Comparison of lineage-specific gene expression revealed similar conservation of gene expression patterns. We also observed that organism-specific cell lineage-associated variation in H3K27 acetylation correlated with organism-specific gene expression changes (*Figure 2D,E* and *Figure 2—figure supplement 1F*). The latter result is consistent with organism-specific regulatory elements maintaining functional impact on Treg cell lineage-specific epigenetic activity.

We next explored conservation of the regulatory network architecture by considering a broader definition of locus epigenetic 'conservation'. Namely, previous studies have identified broad conservation of the regulatory networks even if the exact regulatory elements and transcription factor-binding sites are not conserved (*Stergachis et al., 2014*; *Vierstra et al., 2014*). Thus, we examined all regulatory elements associated with a given gene (most proximal gene body) to see if a lineage-specific regulatory element in mouse could 're-emerge' in a different location as a non-homologous lineage-specific regulatory element in human. This definition could be considered 'functional homology' in contrast to our previous analyses that have focused on individually conserved regulatory elements. Interestingly, this and other functional definitions of lineage-specific conservation (see 'Materials and methods') yielded very few additional genes (statistically non-significant gene counts, see 'Materials and methods'), nearly all of which had non-conserved gene expression patterns in Treg vs conventional CD4[+] T cells (*Figure 2—figure supplement 2*). This implies that lineage specification of Treg cells is governed by a specific evolutionarily constrained small set of regulatory elements, which is in contrast to the more plastic conservation of global genome-wide regulatory network architectures (*Stergachis et al., 2014*; *Vierstra et al., 2014*).

## Conserved binding of the Treg lineage specification factor Foxp3 contributes to the conserved lineage-specific epigenome

Given the critical role of Foxp3 in establishment and maintenance of Treg cell identity and function, and Treg-specific chromatin repression (*Arvey et al., 2014*), we wanted to know if conservation of the

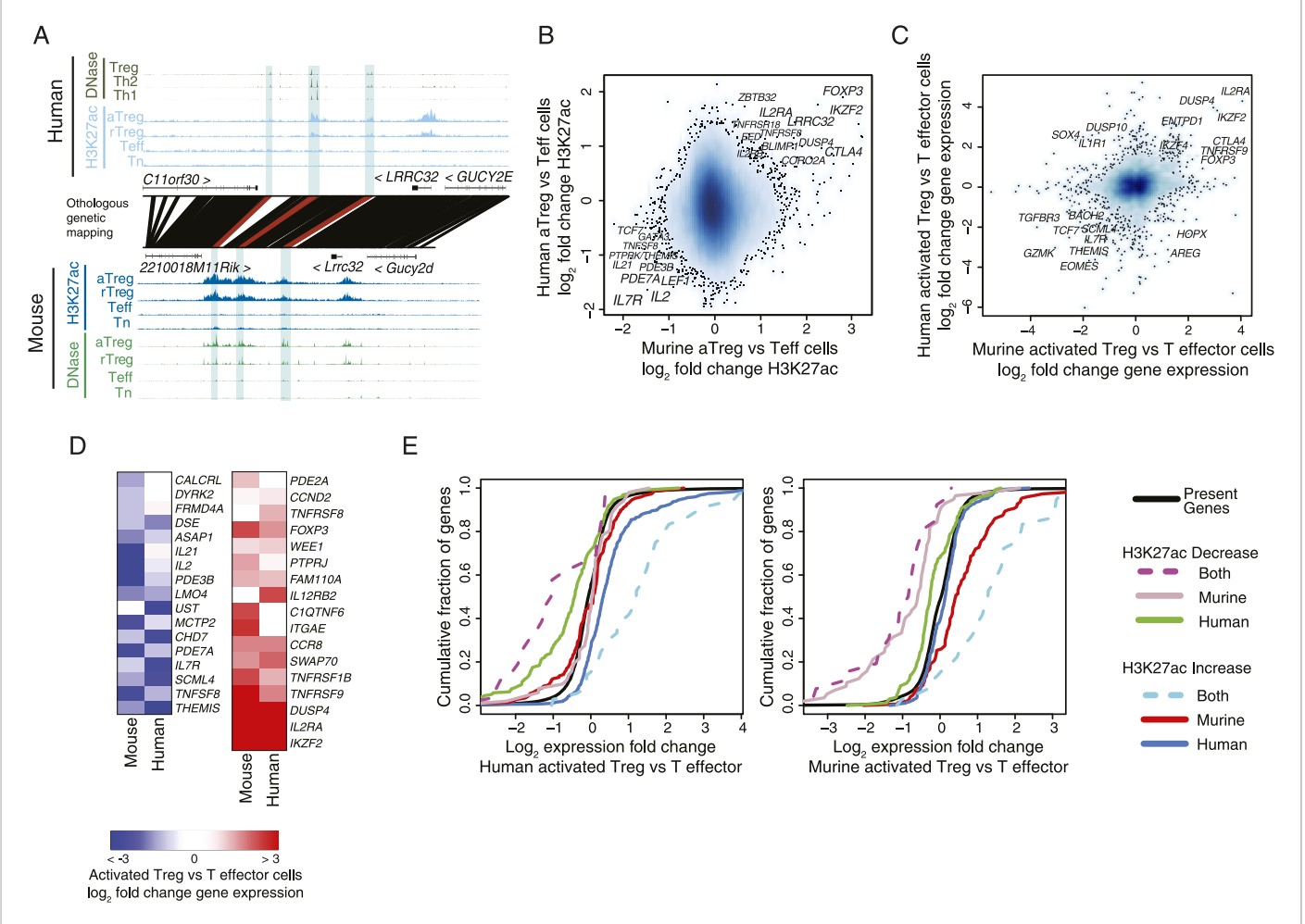

**Figure 2**. Conservation of the lineage-specific epigenome identifies the core Treg cell transcriptional control program. (**A**) The *LRRC32* locus contains multiple epigenetically active regulatory elements that are conserved and Treg lineage-specific in mouse and human. The layout is same as in *Figure 1D,E*, with H3K27ac and DNase-seq RPM quantification shown on the y-axis for multiple mouse and human CD4+ T-cell subpopulations. The DNase-seq track provides high-resolution localization of protein-bound DNA in acetylated loci. (**B**) Lineage-specific conservation of epigenetic activity occurs at only a handful of Treg regulatory elements. Changes in histone acetylation (H3K27ac ChIP RPM) in aTreg compared to Teff cells are shown for mouse (x-axis) and human (y-axis) cells for genetically conserved loci. (**C**) Treg cell lineage-specific gene expression is similarly conserved as lineage-specific chromatin features in mouse and man. Changes in gene expression in aTreg compared to Teff cells are shown for mouse (x-axis) and human (y-axis) cells. (**D**) Conserved gene expression changes are associated with conserved lineage-specificity of histone acetylation. Expression changes are shown for genes most proximal to regulatory elements that were lineage-specific in both species. See also *Figure 2—figure supplement 2*. (**E**) Species-specific gene expression changes are associated with species-specific changes in histone acetylation. Acetylated loci were classified as upregulated or downregulated in a cell type-specific manner in mouse, human, or both organisms. These loci were mapped to their most proximal gene expressed in either mouse or human (lines) and cumulative fraction is shown (y-axis). The differential expression in aTreg vs Teff cells was plotted for these genes in human (left) and mouse (right). Regulatory elements that are lineage-specific in both species (dashed lines) have been associated with genes with the most conserved differential expression patterns. Regulatory elements that are lineage-specific in only mouse (red/gray for up/downregulated loci) are near genes that are more differentially expressed in mouse cell subsets. Similarly, elements that are lineage-specific in only human (purple/green for up/downregulated loci) are associated with differential expression in human. Statistical significance was assessed by the Kolmogorov–Smirnov test; all relevant distribution shifts are statistically significant; p-values are shown for relevant comparisons in *Figure 2—figure supplement 1F*.

The following figure supplements are available for figure 2:

**Figure supplement 1**. The core Treg lineage specification program is robust across multiple donors and is enriched for significant changes in gene expression.

**Figure supplement 2**. Genes with lineage-specific elements in each organism at different genetic locations exhibit limited lineage-specific expression changes.

lineage-specific epigenome is associated with conserved binding sites of Foxp3. We quantified Foxp3 binding in human and murine Treg cells and found that many Foxp3-bound loci were genetically conserved and bound in both organisms (*Figure 3A*, *Figure 3—figure supplement 1A,B*; *Supplementary files 7-9*). While there was substantial conservation of Foxp3 binding, there were also many sites that were species specific or had insufficient binding for robust quantification in the weaker Foxp3-bound sites in one of the species (*Figure 3B*).

Recent reports have implicated Foxp3 in repression of chromatin (*Arvey et al., 2014*; *DuPage et al., 2015*); therefore, we characterized the relationship between conservation of Foxp3 binding and conservation of Treg lineage-specific decreases in H3K27ac. We found that conserved Foxp3 binding was associated with decreased H3K27ac in both human and mouse (*Figure 3C,D*, *Figure 3—figure supplement 1C*). Importantly, at human- or mouse species-specific Foxp3-bound

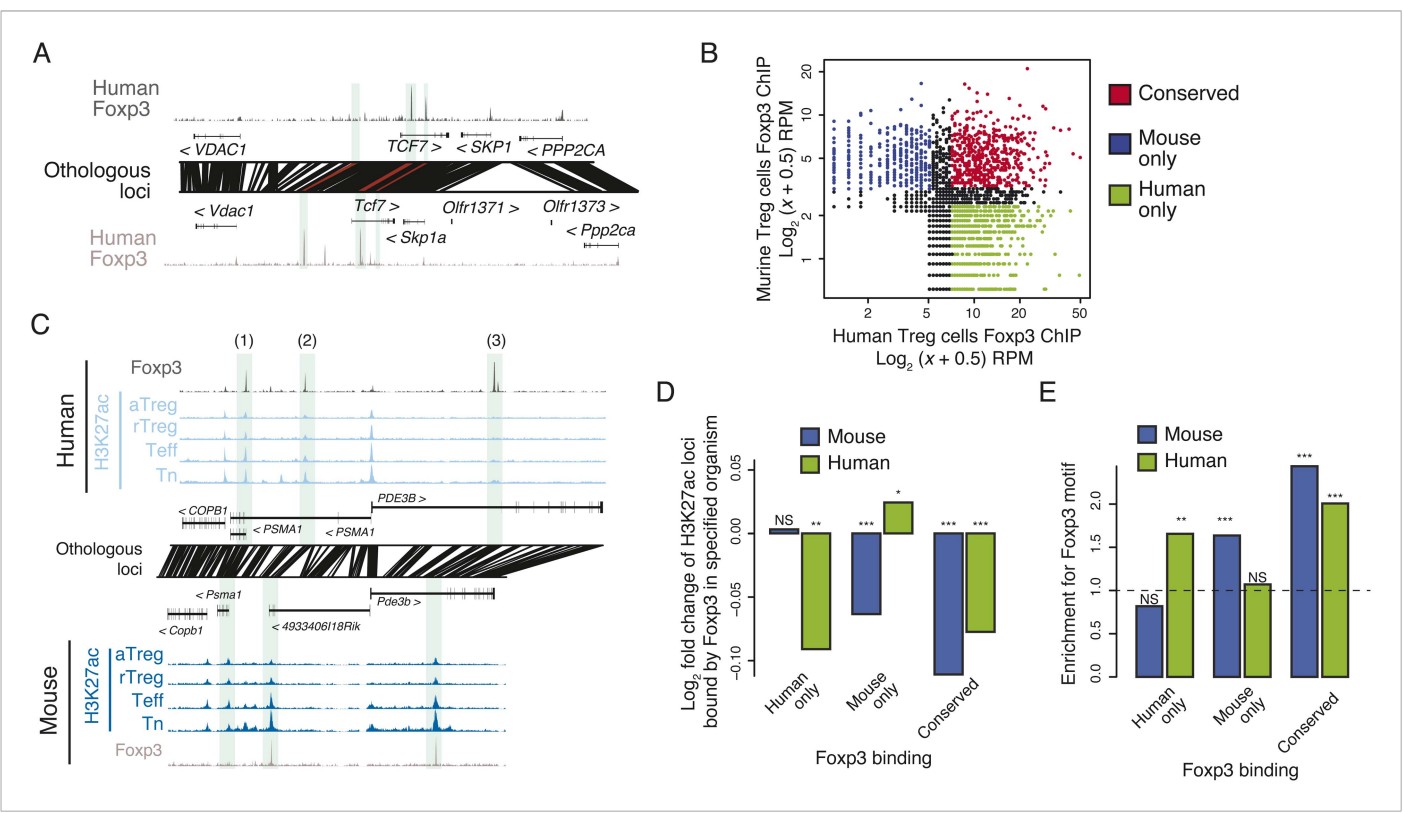

**Figure 3**. Epigenetic and genetic conservation of Foxp3-binding elements and corresponding differential gene expression. (**A**) The *TCF7* locus provides examples of conserved (first intron, promoter-proximal), mouse-specific (upstream), and human-specific (first intron, promoter-distal) Foxp3 binding. Plot layout and axes are similar to *Figure 2A*. Orthologous regulatory elements (light blue) and their orthological mapping (red) are shown. (**B**) Genome-wide, regulatory elements are bound by Foxp3 in both mouse and human (red) or in a human-specific (green) or mouse-specific (blue) manner. (**C**) Foxp3 binds loci in human and mouse that have decreased acetylation in the Treg cells at the *PDE3B* locus. This locus contains species-specific Foxp3-binding sites that are associated with species-specific decrease in acetylation in Treg cells (marked as 1, 3) and conserved Foxp3 binding that is associated with conserved Treg lineage decreases in acetylation (2). The decrease in acetylation at the *PDE3B* promoter is conserved in both species. (**D**) Conserved and species-specific Foxp3 binding is associated with decreased histone H3K27 acetylation at regulatory elements. Changes in human or mouse H3K27 acetylation (y-axis) at genetically conserved regulatory elements are shown to be associated with human- or mouse-specific, or conserved Foxp3 binding (x-axis). Statistical significance was calculated by two-sample t-test using all acetylated loci as the background distribution; p-values are shown as *: $p < 0.05$; **: $p < 0.01$; ***: $p < 0.001$. (**E**) Species-specific Foxp3 binding in mouse and human Treg cells is associated with non-conserved forkhead box DNA motif. Enrichment of the forkhead motif (odds ratio, y-axis) is shown for those Foxp3-binding sites that are genetically conserved and bound only in human (left) or mouse (center) Treg cells or both (right). Odds ratio and statistical significance were calculated using Fisher's exact test, where motif counts in Foxp3-binding sites were compared to flanking 200nt regions to estimate the empirical enrichment over chance. p-values are shown as *: $p < 0.05$; **: $p < 0.01$; ***: $p < 0.001$.

The following figure supplement is available for figure 3:

**Figure supplement 1**. Foxp3 binding sites in mouse and human are reproducible and associated with transcriptional repression.

sites, H3K27ac was decreased exclusively in the corresponding organism (*Figure 3D*, *Figure 3—figure supplement 1D*). Species-specific Foxp3 occupancy at genetically conserved loci was also associated with presence of forkhead DNA-binding motifs at the corresponding Foxp3-bound sites (*Figure 3E*). These results suggest that preservation of the forkhead motif likely enables Foxp3 protein recruitment, Foxp3-mediated chromatin repression at conserved loci, and that DNA sequence divergence can account for differential Foxp3 occupancy of otherwise genetically conserved orthologous regulatory elements.

## Natural human genetic variation at Treg cell lineage-specific regulatory elements is associated with epigenetic variation

To determine if non-coding regulatory elements specific to resting or activated Treg cells are subject to genetic variation in human, we explored polymorphisms associated with these elements. We examined a narrow genomic window (150 bp) at each of the ~85,000 DHSs to find that ~80,000 contained at least 1 polymorphism cataloged in the single nucleotide polymorphism database (dbSNP) and that ~45,000 of these contained polymorphisms with significant (>0.05) minor allele frequency (MAF) in at least one of the thousand genome project (1000G) populations (*Figure 4A*; 'Materials and methods'; *Supplementary file 13*). Consistent with evolutionary constraint, genetically conserved elements had decreased maximum MAF relative to expected frequency (*Figure 4B*; 'Materials and methods'). This pattern held across regulatory elements regardless of their distance from protein-coding regions or binding of Foxp3 (*Figure 4—figure supplement 1A–C*).

To test if genetic variation in the Treg cell-specific regulatory elements could alter enhancer functionality, we characterized DNA sequence polymorphisms associated with inter-individual quantitative variability in H3K27ac modifications across multiple cell populations in a cohort of unrelated donors (*Figure 4—figure supplement 1D,E*). For example, the Treg cell lineage-specific enhancer containing the single nucleotide polymorphism (SNP) rs2882971 at the *ENTPD1* locus demonstrated genetic association with H3K27ac levels across individuals and in an allele-specific manner in a given individual (*Figure 4C*). This is consistent with previous reports showing *ENTPD1* RNA and protein expression association with a strong cis-eQTL (*Orrù et al., 2013*; *Ferraro et al., 2014*). We confirmed the genetic association with chromatin state through meta-analysis of lymphoblastoid cell lines, which enabled sufficient statistical power (*Figure 4D*). We also identified potential associations where the enhancer decorated with H3K27ac was present in multiple CD4$^+$ T-cell subsets, but exhibits only allele-specific variation in Treg cells, suggesting that Treg cell-specific chromatin modulators or transcription factors may preferentially act on a single allele of a polymorphic locus (*Figure 4—figure supplement 1F*).

## Treg cell lineage-specific regulatory elements are enriched for disease-associated polymorphisms

Our observation that the inter-individual genetic variation present in the Treg cell-specific epigenome can potentially alter Treg cell function raised a question if this variation may contribute to polygenic human disease pathogenesis. In particular, while fatal monogenic IPEX disorder resulting from loss-of-function Foxp3 mutations (*Khattri et al., 2003*) highlights the critical role for Treg cells in preventing autoimmunity, the role of these cells in common polygenic human diseases has been notoriously difficult to assess with cellular functional analyses confounded by patients' disease state. As a result, assessments of Treg cell function in patients with complex autoimmune and inflammatory diseases frequently lead to conflicting conclusions open to alternative interpretations (*Miyara et al., 2011*). Furthermore, recent findings from genome-wide association (GWA) studies have revealed that most disease-predisposing susceptibility loci reside in non-coding portions of the genome (*Welter et al., 2014*; *Figure 5—figure supplement 1*), suggesting that the majority of polygenic disease is in fact due to variation in transcriptional control rather than protein structure and function. Recent studies have also found that active chromatin is enriched for variants that contribute to polygenic diseases, including autoimmunity (*Maurano et al., 2012*; *Trynka et al., 2013*; *Farh et al., 2014*), and that these variants can alter transcriptional regulator protein-DNA interactions, resulting in differential gene expression (*Ferraro et al., 2014*; *Lee et al., 2014*; *Ye et al., 2014*).

Therefore, we reasoned that if disease-predisposing polymorphisms were embedded in chromatin with function exclusively in Treg cells, then Treg cell dysfunction could contribute to disease etiology.

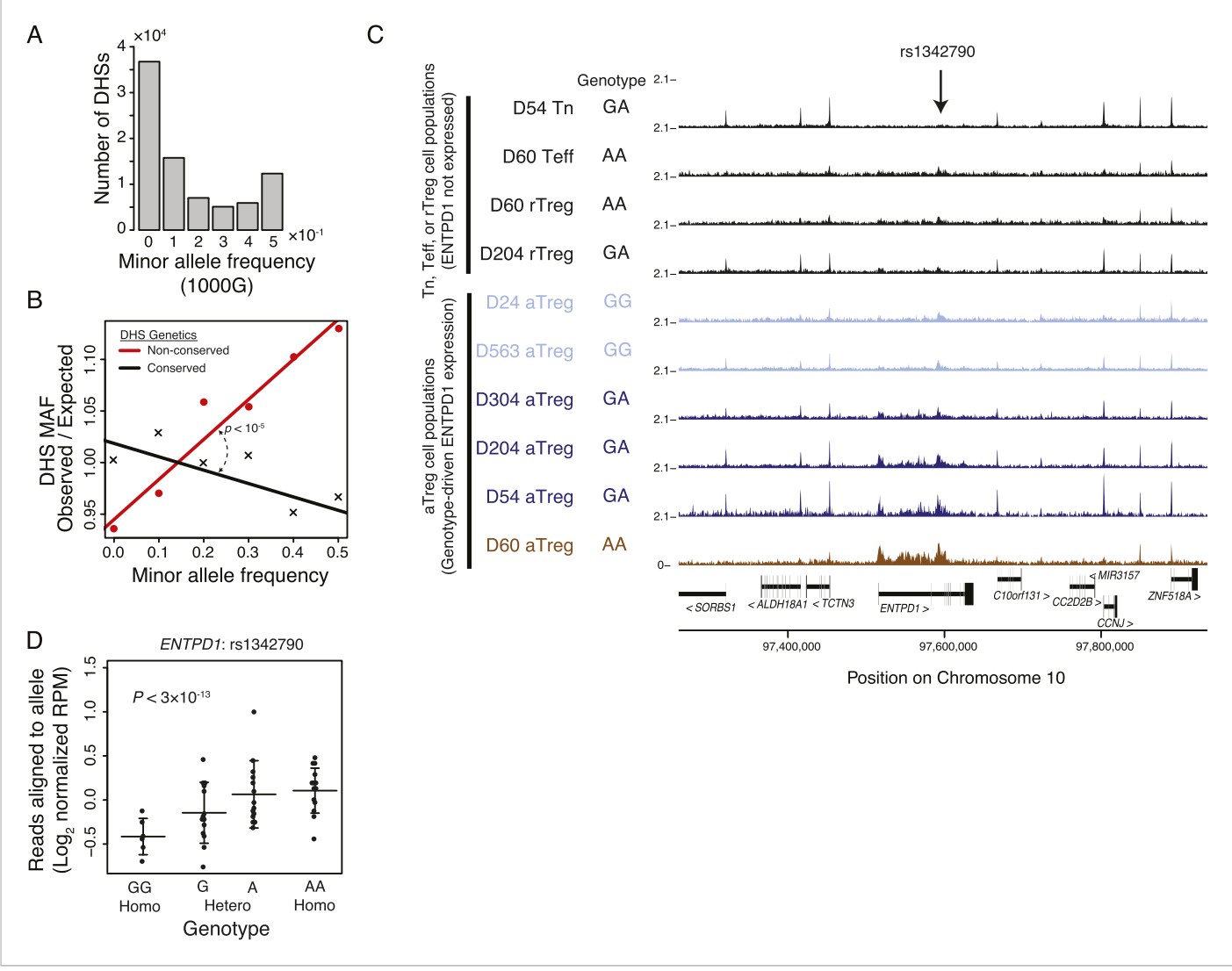

**Figure 4**. Common human genetic polymorphisms in Treg lineage-specific regulatory elements can alter histone acetylation. (**A**) The CD4[+] T-cell epigenome is subject to human-to-human genetic variation. The number (y-axis) of DHSs with at least one polymorphism greater than a given minor allele frequency (MAF, x-axis). MAFs are binned by <0.05 (0), 0.05–0.15 (0.1), 0.15–0.25 (0.2), 0.25–0.35 (0.3), 0.35–0.45 (0.4), >0.45 (0.5) for each population in the 1000G project ('Materials and methods'). (**B**) Human DHSs not genetically conserved in mouse (red) contain greater genetic variation than those DHS that are conserved (black). Empirical p-value is computed by sampling maximum polymorphism MAF across conservation-permuted DHSs, fitting a linear regression, and testing for greater conserved vs shuffled DHS slopes: $\beta_{conserved} > \beta_{shuffle}$. (**C**) Genetic polymorphisms are correlated with H3K27 acetylation at the *ENTPD1* locus. The activated Treg cell-specific expression of ENTPD1 is associated with a haplotype represented by single nucleotide polymorphism (SNP) rs1342790. Resting Treg (rTreg) and naive (Tn) and effector (Teff) CD4[+] T-cell populations have limited acetylation of the *ENTPD1* locus; in contrast, activated Treg cells have acetylation that is associated with the 'A' genotype of rs1342790. (**D**) Quantification of the ENTPD1 genetic association reveals allele-specific histone modification in heterozygous individuals. Meta-analysis of the ENTPD1 locus in B lymphoblastoid cell lines was performed to provide additional statistical power (*Kasowski et al., 2013a*). Genotypes (x-axis) and their corresponding reads crossing the polymorphism per million reads aligned (RPM; y-axis) were calculated for each individual with more than 5 reads at rs1342790. Values across zygosity and cell type were made comparable by doubling heterozygous RPMs and mean normalizing. Statistical significance was estimated by independent assessment of heterozygous and homozygous allelic normalized read counts ('Materials and methods').

The following figure supplement is available for figure 4:

**Figure supplement 1**. Common human genetic polymorphisms exist in Treg lineage-specific regulatory elements and Foxp3 binding sites. These polymorphisms can be associated with variation in histone acetylation.

Through meta-analysis of GWA studies of SNPs, we identified hundreds of statistically significant disease-associated polymorphisms residing in the CD4+ T effector and Treg cell epigenome.

One such strictly Treg cell-specific epigenetic element was found at the *CTLA4* locus, which harbors a risk allele for multiple autoimmune disorders, including rheumatoid arthritis, type 1 diabetes (T1D), Graves' disease, and systemic lupus erythematosus (*Scalapino and Daikh, 2008*). While early GWA genotyping approaches were too coarse to localize the causative polymorphism, fine mapping by the ImmunoChip across a large case–control T1D cohort demonstrated that the most disease-associated variants, represented by rs2882971, lie within Treg cell-specific enhancers that are epigenetically conserved and lineage-specific in both human and mouse (*Figure 5A*, *Figure 2—figure supplement 1A*) (*Onengut-Gumuscu et al., 2015*). Functional relevance of a non-coding *CTLA4* polymorphism in linkage disequilibrium with rs2882971 (rs3087243) has been experimentally validated as influencing the CTLA4 splice form dosage, with disease-predisposing variants decreasing the soluble isoform that is expressed in Treg cells (*Ueda et al., 2003*; *Atabani et al., 2005*; *Gerold et al., 2011*). The linkage disequilibrium across the locus suggests that the causative polymorphism is either in (1) upstream Treg lineage-specific epigenetically active enhancers near rs2882971 or (2) downstream epigenetically and transcriptionally inactive DNA near rs3087243. Pairwise conditioning of rs2882971 and rs3087243 resulted in a non-statistically significant association for both polymorphisms (logistic regression for an additive model yielded SNP coefficient p-values that were greater than 0.05). This indicates that genotype–phenotype statistics alone is unable to resolve the causative SNP. In contrast, our epigenetic approach provides orthogonal data that implicate upstream enhancers.

Next, we sought to identify specific diseases influenced by polymorphisms that may cause Treg cell dysfunction. We analyzed GWA case–control cohorts for autoimmune and autoinflammatory diseases in addition to metabolic and psychiatric disorders ('Materials and methods'; *Supplementary file 14*). We identified dozens of Treg cell lineage-specific epigenetic elements that harbor genetic variation associated with and potentially contributing to polygenic autoimmune disease such as T1D (*Figure 5B,C*; 'Materials and methods'). Analysis of these risk alleles revealed that many were in LD or were proximal to genes with epigenetically conserved Treg lineage-specific elements (*Figure 5D*), suggesting that these elements have important conserved function. In contrast, an aggregate GWAS data set of metabolic (e.g., type 2 diabetes) and psychiatric disorders lacked disease associations in conserved Treg lineage-specific elements. Furthermore, psychiatric disorders risk-alleles were generally localized in non-conserved genetic elements, which is consistent with recent reports finding substantial human-specific genetic diversity in cognition-associated genes in comparison to recent evolutionary ancestors (*Ogawa and Vallender, 2014*).

## Discussion

In the past few years, many studies have characterized genome-wide epigenetic activity in progenitor cells or related differentiated cell lineages to identify hundreds of lineage-specific regulatory elements (*Epigenome Roadmap Consortium, 2015*). To isolate biologically important elements, conservation of genetic DNA sequence and epigenetic activity has been used as a proxy for functional significance (*Odom et al., 2007*; *Schmidt et al., 2010*; *Long et al., 2013*; *Vierstra et al., 2014*). We compared genome-wide features of regulatory elements and their activity in two closely related cell lineages, Treg cells, and effector CD4+ T cells, which represent alternative cell fate choices during T-cell differentiation and fulfill opposing anti-inflammatory and pro-inflammatory immune functions. In our comparison of Treg and effector T-cell lineages, we found that ~85% of regulatory elements were genetically conserved and epigenetically active in both human and mouse, ~1–2% were lineage-specific in human or mouse, and <0.1% were lineage-specific in both human and mouse. This indicates that of those elements that are specific to the Treg lineage, less than 10% were lineage-specific in both mouse and human. Thus, the bulk of lineage-specific epigenetic activity was unique to each organism with only a small set of conserved Treg lineage defining regulatory elements, including enhancers near *FOXP3*, *IL2RA*, *LRRC32*, *IKZF2*, *IL2*, *IL7R*, and *PDE3B*. Interestingly, our analysis also indicates the potential importance of *CCR8*, *DUSP4*, *TNFRSF19*, *TNFRSF1B*, *THEMIS*, *PDE7A*, and *SWAP70* to Treg cellular function, as lineage-specific epigenetic activity was conserved in both mouse and human. These findings suggest that lineage specification programs may be dependent on but a few immutable genetic regulatory elements, while tolerating considerable epigenetic variation at numerous additional organism-specific lineage-specific elements.

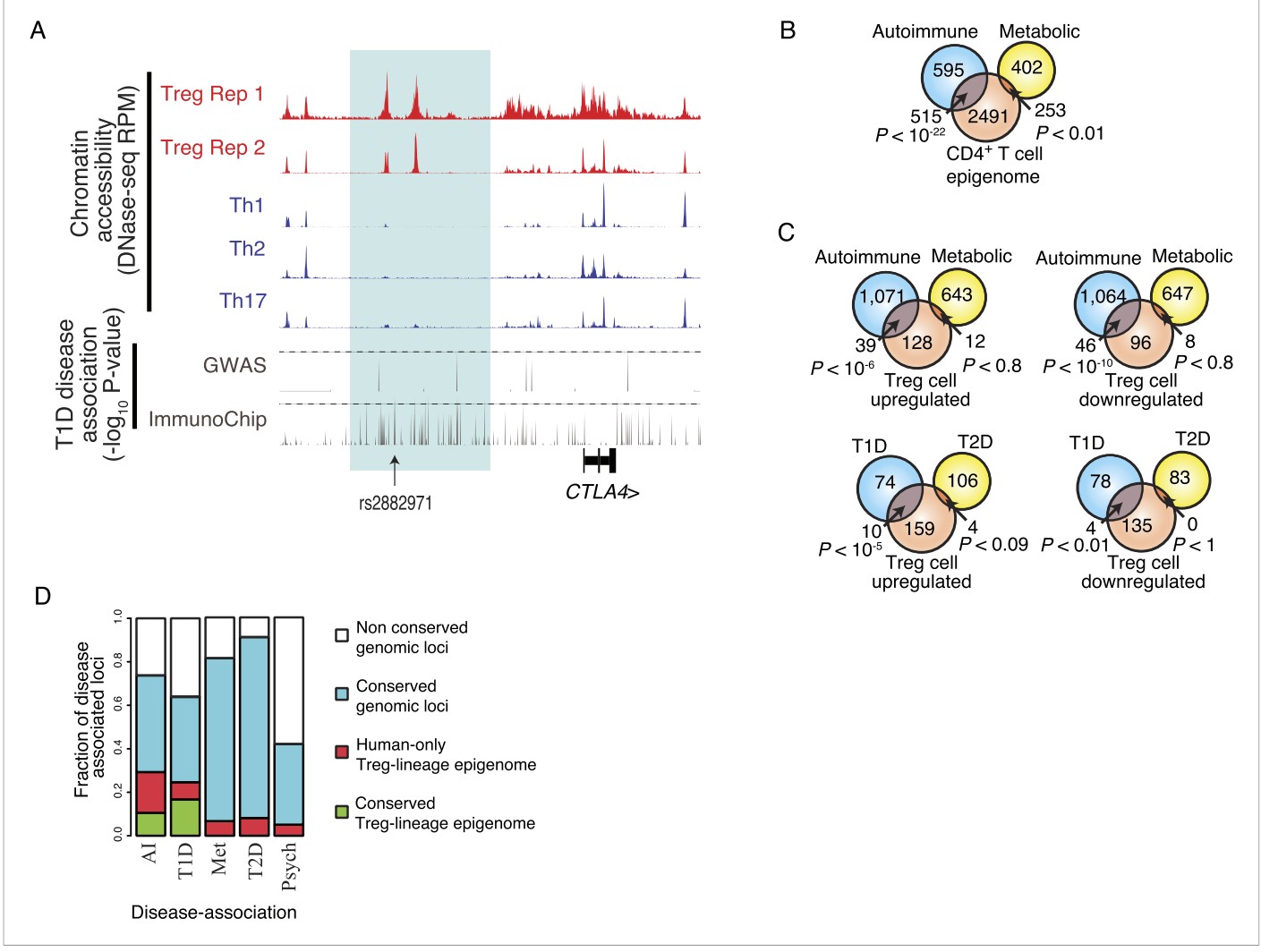

**Figure 5**. Disease-associated SNPs are enriched in Treg lineage-specific regulatory elements. (**A**) Fine mapping of genetic disease association identifies Treg-specific upstream enhancers (highlighted in blue) at the *CTLA4* locus as being associated with predisposition to type 1 diabetes (T1D). The high-resolution ImmunoChip SNP array analysis suggests that the functional polymorphism resides in the Treg-specific enhancer. The x-axis shows genomic position and y-axis shows RPM for chromatin tracks and −log10(p-value) for association study data. Dashed horizontal lines show genome-wide statistical significance thresholds for T1D disease association studies. The highlighted polymorphism rs2882971 is representative of multiple SNPs in linkage disequilibrium. (**B**) Autoimmune disease-risk polymorphisms have extensive overlap with the pan-CD4+ T-cell epigenome. Disease-risk polymorphisms was obtained from the NHGRI GWAS catalog and statistical significance of overlap was determined by a one-tailed hypergeometric test using all known disease-associated polymorphisms to model the null distribution ('Materials and methods'). (**C**) Treg and Teff cell-specific epigenetic elements contain a significant number of autoimmune disease-risk polymorphisms. Metabolic diseases are shown as a control. T1D and type 2 diabetes (T2D) are shown as representative autoimmune and metabolic diseases. Aggregated disease sets are provided in *Supplementary file 14*. (**D**) Polymorphisms in conserved Treg lineage-specific epigenetic elements are enriched for autoimmune-associated genetic variation. Genes near epigenetic elements containing risk polymorphisms are divided into categories: genetically non-conserved, genetically conserved, lineage-specific elements epigenetically active in human, and lineage-specific epigenetic modification conserved in both human and mouse. Autoimmune (AI), T1D, metabolic (Met), T2D, and psychiatric (Psych) disease sets are shown.

The following figure supplement is available for figure 5:

**Figure supplement 1**. Non-coding disease-associated polymorphisms are assayed by genotyping arrays.

Given previous findings regarding the extensive turnover of DNA elements in genome-wide regulatory networks (*Stergachis et al., 2014*; *Vierstra et al., 2014*), it is noteworthy that we found limited turnover of regulatory elements that are lineage-specific in both mouse and human. This

confirms the unique importance of these individual regulatory elements. However, our observation that many lineage-specific regulatory elements are lineage-specific in only a single organism highlights the plasticity of genome-wide regulatory networks. Furthermore, we found that the Foxp3-mediated transcriptional control program was subject to forkhead-binding motif 'turnover' that was associated with functional consequences. While these organism-specific elements contributed to gene regulation, the regulatory elements with conserved lineage-specific epigenetic activity were associated with far more pronounced regulation of Treg cell lineage-specific gene expression.

Although our studies support a role for Treg cell dysfunction in disease pathogenesis, other factors are almost certainly required, such as environmental contributors and dysfunction of multiple other cell lineages and physiological processes. For instance, it is reasonable to consider that autoimmune diabetes development is impacted most significantly by HLA risk alleles (*Wong et al., 2004*; *Stadinski et al., 2010*), and to a lesser degree by other susceptibility determinants including potentially compromised Treg cells. Though, we observed statistical enrichment of risk alleles near Treg cell enhancers, it is worth noting that the majority of polymorphisms exist as broad haplotype blocks that cannot be perfectly resolved to a single or several lineage-specific enhancers. Although methods for parsing risk alleles into causative and specific cell lineages have been proposed (*Maurano et al., 2012*; *Trynka et al., 2013*; *Farh et al., 2014*; *Seumois et al., 2014*; *Onengut-Gumuscu et al., 2015*), these approaches, similar to our study, rely on aggregate proximity statistics and imperfect modeling of genetic linkage. As whole genome sequencing and mappings of epigenetic activity improve and expand to more cell populations (*Epigenome Roadmap Consortium, 2015*), we expect a significant fraction of loci will have clear epigenetic partitioning in many diseases. We propose that for those risk alleles without clear causative polymorphism due to broad linkage disequilibrium, hypothesis-driven lineage-focused epigenomic QTL studies may be an alternative for gaining increased resolution of disease causing polymorphisms.

Our study suggests that the vast majority of Treg cell lineage-specific elements are not conserved between mice and humans suggesting that during cellular differentiation, only a handful of epigenetic elements implement the core transcriptional program that underlies establishment of a differentiated Treg cell identity and function. In support of this model, disease-associated polymorphisms clustered at genic loci containing core lineage-specific epigenetic elements. Nevertheless, most Treg lineage-specific regulatory elements were lineage-specific only in human and could harbor extensive natural genetic variation that could influence transcription, alter cellular function, and contribute to polygenic disease. This orthogonal high-resolution epigenetic and genetic analysis enabled characterization of disease-predisposing genetic variation associated with Treg-specific enhancers, and therefore, implicated Treg cells in complex autoimmune disease.

## Materials and methods

### Cell isolation

Buffy coat preparations from normal human peripheral blood donors were obtained from the New York Blood Center (NYBC) on same day as donation. CD4$^+$ T cells were enriched through negative selection with RosetteSep antibody cocktails (Stem Cell Technologies #15062, Vancouver, BC Canada). Isolated CD4$^+$ T cells were stained for CD3 (PE-TexasRed, Invitrogen #MHCD0317, Waltham, MA, United States), CD4 (APC-eFluor 780, eBioscience #47-0049-42, San Diego, CA, United States), CD45RA (APC, BioLegend #304112), CD45RO (PE-Cy7, BioLegend #304230), and CD25 (PE, BioLegend #302606), and resting and activated Treg and effector CD4$^+$ T-cell subsets were purified on a BD Biosciences Aria2 fluorescent cell sorter.

Murine CD3$^+$CD4$^+$ cells were isolated as previously described (*Kim et al., 2007*; *Arvey et al., 2014*). Briefly, naive CD4$^+$ T cells (GFP$^-$) and resting Treg cells (GFP$^+$) were isolated from untreated *Foxp3*$^{DTR}$ mice, whose Treg cells express GFP-DTR (DT Receptor) fusion protein driven cells by an IRES-DTR-GFP coding DNA sequence knocked into 3′-UTR of the *Foxp3* gene. Activated T effector cells (GFP$^-$) and activated Treg cells (GFP$^+$) were FACS sorted from *Foxp3*$^{DTR}$ mice 11 days after DT treatment (day 0 and 1; 20 µg/kg). DT treatment resulted in transient ablation of Treg cells and systemic inflammation. Cells harvested from lymph node and spleen were enriched by positive selection (Dynabeads, Invitrogen) and sorted on a FACS Aria2 fluorescent sorter.

## ChIP-seq analysis

ChIP-seq analysis of H3K27Ac in mouse and human cells was performed as previously described (*Arvey et al., 2014*). Briefly, ChIP was performed using the H3K27Ac-specific antibody (Abcam ab4729, Cambridge, United Kingdom) raised against the conserved acetylated epitope 'LATKAARK-SAPA'. Sequencing was performed using an Illumina Hi-Seq 2000 instrument and standard library preparation and adaptors. Reads were aligned to hg19 using BWA using the 'mem' algorithm with parameters '-k 22 -L'. Only nuclear chromosomes were included in our analysis, which excluded all contigs of unknown physical mapping, episomal DNA, and mitochondrial DNA. Peaks were called using MACSv2 using default parameters and a permissive threshold for total peak count (p < 0.0001). The peaks for all experiments were then combined and reads from all experiments were remapped onto the combined set of peaks. We removed peaks with low reads per million (<1 RPM in all samples) or high-input signal (>0.5 RPM); or peaks that were blacklisted by the ENCODE consortium analysis of artifactual signals in mouse or human cells (*Kundaje, 2013*). ChIP tracks were plotted by taking strand-shifted reads (covering 200 nt from original read start) and computing overlaps for the center 100 nt of each read, as previously described (*Arvey et al., 2014*). This simple transformation is superior to simple raw read overlap, as it enhances signal at actual protein–DNA-binding sites.

## Meta-analysis of epigenetic and gene expression data sets

Human and murine T effector and Treg cell DHSs were assayed as previously described (*Bernstein et al., 2012*; *Samstein et al., 2012*). To confirm specificity of the most Treg-specific regulatory elements, we compared active regulatory elements across a number of tissues and immune cell populations identified by ChIP-seq and DNase-seq by the Epigenome Roadmap and ENCODE consortia (*Bernstein et al., 2012*). The pancreatic islet active genomic element data set was obtained from recent studies (*Pasquali et al., 2014*; *Epigenome Roadmap Consortium, 2015*). Additional human Treg cell epigenetic data were obtained from recent studies (*Andersson et al., 2014*; *Schmidl et al., 2014*). Murine Foxp3 ChIP-seq data sets were generated in our previous studies (*Samstein et al., 2012*; *Arvey et al., 2014*) and human Foxp3 ChIP-seq data sets were obtained from SRP006674 (technical replicates SRR192544, SRR192545) and SRP017669 (SRR639419, SRR639420) (*Birzele et al., 2011*; *Schmidl et al., 2014*). Reproducibility of mouse and human ChIP-seq studies was confirmed (*Figure 3—figure supplement 1*). Murine gene expression microarray data sets were generated in our previous studies (*Samstein et al., 2012*; *Arvey et al., 2014*). Mouse *Foxp3* gene expression is derived from RNA-seq (data not shown) since the Affymetrix 430 2.0 array probe assays the 3′-UTR region of the gene that contains the knocked in IRES-DTR-GFP construct. Human gene expression was obtained from a previously published study (*Miyara et al., 2009*).

## Quantifying genetically and epigenetically conserved loci

Acetylated, DNase hypersensitive, and Foxp3-bound loci were identified as described above. To identify orthologous loci in human and mouse, the UCSC liftOver program (*Hinrichs et al., 2006*) was used with the hg19 to mm9 chains downloaded from: human: http://hgdownload-test.cse.ucsc.edu/goldenPath/hg19/liftOver/, mouse: http://hgdownload-test.cse.ucsc.edu/goldenPath/mm9/liftOver/ using default parameters. If an orthologous locus was not identified, iterations over smaller regions of the locus were tested for orthology. This was particularly important for acetylation events, which typically spanned 2–5 kbp, with only a fraction requiring conservation to obtain conserved function. We used the following iteration scheme, starting at the center and then scanning outward in increments depending on assay: histone acetylation ChIP-seq: 3–5 kbp loci were searched in 300 bp increments, DNase-seq: 200 bp loci were searched in 50-bp increments, Foxp3 ChIP-seq: 300 bp loci were searched in 50-bp increments.

Our high-level results were robust to diverse parameter settings and code is provided online. Conserved epigenetic activity was quantified as follows. After matching human-to-mouse loci, we then mapped all mouse-to-human elements and took the union of all elements in each species, respectively. Raw read counts were mapped to the union set of all genetically conserved regulatory elements and non-genetically conserved loci were set to zero. For analysis of gene-level conservation (e.g., gene expression), human HUGO gene names were mapped to murine gene names by Mouse ENCODE mapping (*Yue et al., 2014*).

## Quantifying mobility and function of lineage-specific regulatory elements

To test for mobility (since individual regulatory elements may be active in both mouse and human, but be lineage-specific only in a single organism, we determined if lineage-specific regulatory elements appear as an independent regulatory elements near the same gene) of lineage-specific epigenetic elements within a genic locus (defined as the closest gene transcription start site (TSS) and genes within 100 kbp), we addressed the following questions: (1) Is there mobility of lineage-specific epigenetic elements within a locus? (2) Do lineage-specific epigenetic elements become epigenetically active, but lineage non-specific, conserved genetic elements? (3) Would loosening of cutoffs for lineage-specificity and taking the maximally lineage-specific element across a large locus reveal conservation of the lineage specification program across non-conserved regulatory elements?

For (1), we identified genic loci that contained movement of lineage-specific epigenetic elements in mouse and human (*Figure 2—figure supplement 2D*, example shown in *Figure 2—figure supplement 2A*). Gene expression was concordant for 3/45 elements examined (not statistically significant).

For (2), a handful of lineage-specific elements became epigenetically inactive in the other organism (<15, not statistically significant) and had limited lineage-specific gene expression (*Figure 2—figure supplement 2E*).

For (3), loosened rank cutoffs identified weakly lineage-specific elements in human (n = 632/496, up/down) and mouse (n = 759/625, up/down) at proximal but not necessarily orthogonal gene loci. Nearby genes were largely unaffected or exhibited discordant lineage-specific gene expression, indicating that this set may have limited functional relevance. Nearby genes with weak levels of differential expression (>0.5-fold or $q < 0.01$, where $q$ is false discovery rate (FDR) value) are shown in *Figure 2—figure supplement 2F* (n = 26 up; n = 22 down), which includes many genes with regulatory elements that are lineage specific in both mouse and human as characterized in *Figure 2—figure supplement 2C*.

## SNP genotype calling

Genotype calling was performed according to best practices described by the Broad Institute Genome Analysis Tool Kit v3.1.1 (GATK) documentation (*DePristo et al., 2011*). We used Picard (v0.92) to designate read groups by donor and cell type. Potentially monoclonal reads were removed by selecting a single read at random for each position in the genome with multiple read start sites (custom Python script). This strategy avoids reference alignment bias of other tools (e.g., samtools rmdup) that select reads with maximum alignment score. The standard GATK pipeline was used to perform local realignment around potential indels and recalibrate quality scores across experiments. Final calling and estimation of allelic read depth were performed using the Unified Genotyper algorithm on variants compiled in dbSNP (v138). Variant calling quality was assessed by the Variant Recalibrator algorithm using HapMap 3.3, 1000G Project, and dbSNP v138 as calibrating resources for estimating sensitivity and specificity of calls. Low-quality tranches were discarded and not considered in downstream analyses.

## Allele-specific chromatin modifications

Heterozygous SNPs were identified using GATK (see above). This set of SNPs was then filtered to remove low quality or unreliable genotype calls by discarding SNPs that:

1. failed QC after GATK quality recalibration (as described above);
2. were heterozygous in less than 2 donors;
3. were homozygous in less than 2 donors;
4. showed allelic imbalance in input ('non-ChIPed') DNA at nominal *p*-values <0.05.

Allelic read depth was determined from GATK estimates from 'rmdup' data output. Two-tailed binomial tests for allelic imbalance was used to filter all heterozygous polymorphisms with p-values > 0.2, as these provide little evidence for allele-specific enhancer usage. Remaining polymorphisms were aggregated into H3K27ac regulatory elements and tested for LD $r^2 > 0.99$ in the five 1000G populations AFR, EUR, CEU, CHB, and JPT. These polymorphisms were then aggregated for each regulatory element, under the assumption of allelic read depth

independence. This statistic was then analyzed for divergence from an empirical null distribution constructed by random uniform p-value aggregation under identical assumptions. This is shown in the QQ-plot of *Figure 4—figure supplement 1E*.

Since B lymphoblastoid cell lines (B-LCL) share many regulatory elements with CD4+ T cells, we performed meta-analysis with the inclusion of the B-LCL data sets (*Kasowski et al., 2013a*; *McVicker et al., 2013a*) to increase confidence in shared allele-specific chromatin expression alleles (e.g., *ENTPD1* as shown in *Figure 4*). Additionally, a small set of loci presented allele-specific modifications in only CD4+ T cells (e.g., *Figure 4—figure supplement 1F*); however, our analysis was underpowered to detect such events genome-wide.

A test of allelic preference across both homozygous and heterozygous individuals was performed on normalized estimates of read count coverage of a given genotype. That is, reads with a given genotype were quantified by RPM. Heterozygous RPM values were doubled to make them comparable to homozygous RPM values. These were then mean-normalized by data set (LCL and primary T cells). Statistical test of allele specificity incorporating homozygous individuals was performed using two approaches: (1) independent test of heterozygous and homozygous individuals and (2) linear regression across all genotypes. For (1), test of heterozygous allele specificity is described above and quantitative differences in H3K27ac across homozygous individuals was estimated by t-test. As homozygous and heterozygous individuals represent independent observations, the product of p-values was used as an estimate of statistical significance of allele specificity. For (2), linear regression of normalized RPM values to major allele counts was modified to include values for the two heterozygous read overlap counts, shifting the regression domain from {0, 1, 2} to {0, 1, 2, 3}, where 0, 3 are homozygous and 1, 2 are the respective heterozygous regressor values.

## Analysis of epigenetic association with disease-risk SNPs

Polymorphisms identified in prior GWA studies were obtained from the NHGRI GWAS Catalog (*Welter et al., 2014*) (downloaded 12 April, 2014). At this time, the catalog was in hg18 coordinates, which were converted to hg19 coordinates using the liftOver program and UCSC whole genome lift over chains. These polymorphisms were then analyzed for overlap with regions surrounding regulatory elements using the hypergeometric overlap test. Our analysis crudely corrected for linkage disequilibrium, which has confounded significance statistics in previous studies (*Maurano et al., 2012*), by taking broad 100 kbp regions (or most proximal gene body) surrounding regulatory elements to include disease-associated polymorphisms across the haplotype block. This empirically eliminated over-counting events arising from multiple epigenetic elements overlapping a single genetic haplotype. We include GWA studies if they identified >5 independent SNPs associated with the disease. The phenotypic traits and diseases analyzed are presented in *Supplementary file 14*.

## Source Code

All source code is open source and available for download at https://bitbucket.org/aarvey/treg_epigen.

## Ethics statement

All mice were bred and housed in the animal facility at the Memorial Sloan Kettering Cancer Center, in accordance with institutional guidelines.

Human buffy coat samples were obtained from NYBC. Human subject informed consent was obtained by the NYBC, under guidance of the NYBC committee for protection of human subjects. The informed consent allowed for research use and publication. Donor identities were anonymous and subject IDs are different from those provided by NYBC.

The Memorial Sloan Kettering Cancer Center IRB determined analysis of human data to be exempt research per 45 CFR 46.101.b(4), 45 CFR 164.512(i)(2)(ii), and 45 CFR 46.116(d).

# Additional information

## Funding

| Funder | Grant reference | Author |
| --- | --- | --- |
| National Institutes of Health (NIH) | 5R37AI034206 | Alexander Y Rudensky |

| Funder | Grant reference | Author |
|---|---|---|
| Howard Hughes Medical Institute (HHMI) | | Alexander Y Rudensky |
| Cancer Research Institute (CRI) | Predoctoral Fellowship | Joris van der Veeken |
| National Cancer Institute (NCI) | P30 CA008748 | Alexander Y Rudensky |
| National Institutes of Health (NIH) | U01 HG007893 | Alexander Y Rudensky |

The funders had no role in study design, data collection and interpretation, or the decision to submit the work for publication.

## Author contributions

AA, AYR, Conception and design, Acquisition of data, Analysis and interpretation of data, Drafting or revising the article, Contributed unpublished essential data or reagents; JV, Conception and design, Acquisition of data; GP, Conception and design, Contributed unpublished essential data or reagents; SSR, PC, Acquisition of data, Contributed unpublished essential data or reagents

## Ethics

Human subjects: This work study was approved by MSKCC IRB, approval number WA0398-12.

# Additional files

## Supplementary files

• Supplementary file 1. Quantification of human H3K27ac peaks. Reads per million reads aligned (RPM) are $\log_2(x + 0.5)$ transformed and quantile normalized.

• Supplementary file 2. Quantification of human DNase-seq peaks. RPM reads aligned are $\log_2(x + 0.5)$ transformed and quantile normalized.

• Supplementary file 3. Quantification of mouse H3K27ac peaks. RPM reads aligned are $\log_2(x + 0.5)$ transformed and quantile normalized.

• Supplementary file 4. Quantification of mouse DNase-seq peaks. RPM reads aligned are $\log_2(x + 0.5)$ transformed and quantile normalized.

• Supplementary file 5. Orthologous regions are determined for the union of all human and mouse acetylated loci. Values are averages across indicated cell populations after quantile normalization.

• Supplementary file 6. Orthologous regions are determined for the union of all human and mouse DNase hypersensitive sites (DHSs). Values are averages across indicated cell populations after quantile normalization.

• Supplementary file 7. Quantification of human Foxp3 ChIP-seq peaks. RPM reads aligned are $\log_2(x + 0.5)$ transformed and quantile normalized.

• Supplementary file 8. Quantification of mouse Foxp3 ChIP-seq peaks. RPM reads aligned are $\log_2(x + 0.5)$ transformed and quantile normalized.

• Supplementary file 9. Orthologous regions are determined for the union of all human and mouse Foxp3 peaks. Values are averages across indicated cell populations after quantile normalization.

• Supplementary file 10. Gene expression of human.

• Supplementary file 11. Gene expression of mouse.

• Supplementary file 12. Gene expression of orthologous genes for uniquely mapping human/mouse genes.

• Supplementary file 13. Maximum minor allele frequency in each DHS with genetic, epigenetic, conservation, and genic localization data.

• Supplementary file 14. Autoimmune and metabolic phenotypes genome-wide association (GWA) studies analyzed.

## Major datasets

The following dataset was generated:

| Author(s) | Year | Dataset title | Dataset ID and/or URL | Database, license, and accessibility information |
|---|---|---|---|---|
| Arvey A, van der Veeken J, Plitas G, Rudensky AY | 2015 | Genetic and epigenetic variation in the lineage specification of regulatory T cells | http://www.ncbi.nlm.nih.gov/geo/query/acc.cgi?acc=GSE73418 | Publicly available at the Gene Expression Omnibus (Accession no: GSE73418). |

The following previously published datasets were used:

| Author(s) | Year | Dataset title | Dataset ID and/or URL | Database, license, and accessibility information |
|---|---|---|---|---|
| ENCODE | 2012 | Encode datasets | http://genome.ucsc.edu/ENCODE | |
| Epigenome Roadmap Consortium | 2015 | Epigenome roadmap datasets | http://www.roadmapepigenomics.org | |
| McVicker G, van de Geijn B, Degner JF, Cain CE, Banovich NE, Raj A, Lewellen N, Myrthil M, Gilad Y, Pritchard JK | 2013 | Identification of genetic variants that affect histone modifications in human cells | http://www.ncbi.nlm.nih.gov/geo/query/acc.cgi?acc=GSE47991 | Publicly available at the Gene Expression Omnibus (Accession no: GSE47991). |
| Kasowski M, Kyriazopoulou-Panagiotopoulou S, Grubert F, Zaugg J, Kundaje A, Liu Y, Boyle AP, Zhang QC, Zakharia F, Spacek DV, Li J, Xie D, Olarerin-George A, Steinmetz LM, Hogenesch JB, Kellis M, Batzoglou S, Snyder M | 2013 | Extensive Variation in Chromatin States Across Humans | http://www.ncbi.nlm.nih.gov/geo/query/acc.cgi?acc=GSE50893 | Publicly available at the Gene Expression Omnibus (Accession no: GSE50893). |
| Kilpinen H, Waszak SM, Gschwind AR, Raghav SK, Witwicki RM, Orioli A, Migliavacca E, Wiederkehr M, Gutierrez-Arcelus M, Panousis N, Yurovsky A, Lappalainen T, Romano-Palumbo L, Planchon A, Bielser D, Bryois J, Padioleau I, Udin G, Thurnheer S, Hacker D, Core LJ, Lis JT, Hernandez N, Reymond A, Deplancke B, Dermitzakis ET | 2013 | ChIP-seq of 14 human lymphoblastoid cell lines from the 1000 Genomes sample set | https://www.ebi.ac.uk/arrayexpress/experiments/E-MTAB-1884/ | Publicly available at Array Express (Accession number E-MTAB-1884). |

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
