## [Decision Letter]

Thank you for sending your work entitled “Genetic and epigenetic variation in the lineage specification of regulatory T cells” for consideration at *eLife*. Your article has been evaluated by Mark McCarthy (Senior editor) and three reviewers, one of whom is a member of our Board of Reviewing Editors.

The manuscript by Arvey et al. explores genetic and epigenetic variation in the lineage specification of regulatory T cells. A comparison is made of various mouse and human T helper cell populations using H3K27ac as a surrogate of active regulatory elements. The studies are of interest because of the key roles of Treg cells in control of immune function and autoimmunity. There were divergent opinions among the reviewers as to the suitability of this manuscript for publication in *eLife* that related to the extent to which it represents a conceptual advance. There were also concerns that many aspects of the manuscript are insufficiently documented and difficult to follow. After considerable discussion among the reviewers, a decision was made to allow consideration of a revised manuscript that fully addresses the main concerns. The consensus was that this could not be accomplished in the form of a short report due to the requirement to provide more detail and that the revised manuscript would have to be expanded to a full article rather than a short report.

Essential revisions:

1) A major motivation of this work is based on the statement, ‘it remains unknown if changes in the chromatin state of regulatory elements associated with cell lineage specification are conserved in their lineage-specificity’. However, this precise question was the major focus of Stergachis et al. Nature 515, 356-70, 2014. The Stergachis et al. paper, which is not cited, systematically analyzed the cis-acting circuitry of a large number of mouse and human cell types. A main point of Stergachis et al., is that while the precise genomic sequences containing cis regulatory elements for key transcription factors may not be conserved, the higher level organization of transcriptional networks is highly conserved. Although not the main focus of the paper, Treg cells are included in this analysis in Figures 3, 4 and 5. The methods differ in that Stergachis et al. primarily rely on DNase hypersensitivity, which indicates TF binding, whereas the current manuscript uses H3K27ac to identify regions of regulatory activity. The current findings may be fully consistent with those of Stergachis, but at this point it is not clear the conclusion in the Abstract that ‘our findings suggest that a small set of regulatory elements specify the Treg lineage’ is correct. This conclusion is drawn from the data shown in Figure 2 and Figure 2—figure supplement 1, and is based on the relatively limited set of ‘conserved’ elements. These studies do not exclude the possibility that there are many non-conserved elements (that may represent conserved functional modules in different genomic locations) that are important for expression of Treg-specific genes. The authors will need to directly compare and contrast their findings and conclusions with those of Stergachis et al. with respect to the point of whether a small set of regulatory elements is sufficient to specify the Treg lineage.

2) Many of the figure panels in this manuscript are hard to interpret, and are inadequately explained in the legends. Specific examples of this are detailed in ‘Additional concerns’. Secondly, the Methods section does not give enough detail about the analyses that were performed to explain how certain numbers were arrived at. There needs to be more detail about what computational processes and parameters were used. Ideally, code would be provided as well to enable thorough review of what is largely a data-analysis, rather than data-generation, paper. Thirdly, many of the claims relied on single examples, and many figures lacked sufficient numerical or statistical underpinnings. Also, there should be more discussion about the relationship between this analysis and the results in [27]; [19] and [62].

3) Figure 2 show that only a few genes show correlation between K27 levels in human and mouse, and only a few genes show correlation in expression between human and mouse. It would seem that a listing of the genes in common between these two analyses would make the analysis much more meaningful.

4) Figure 2 is a very strong demonstration that sites bound by Foxp3 only in human Treg cells lost H3K27ac only in humans, and conversely sites bound by Foxp3 only in mice lost H3K27 only in mice. In conserved sites both human and mice lost H3K27ac. This appears to support the notion that Foxp3 is associated with repression, but there is no comment on this. Would the authors discuss?

5) Figure 3 expounds on the ENTPD1 gene encoding the ectonucleoside triphosphate diphosphohydrolase, CD39. They show and state, “*ENTPD1* locus demonstrated genetic association with H3K27ac levels across individuals and in an allele-specific manner in a given individual (Figure 3).” The odd thing about this is that CD39 is ubiquitously expressed in virtually all tissues. Although the authors show an association with H3K27 levels, they did not show whether this translated into changes in gene expression. I would think the authors should address this issue, especially as CD39 is so widely expressed.

6) Figure 3 shows the K27ac reads associated with two alleles in the same individual, which is a nice analysis; however, do the points represent individual people? How many are people, and how many are lymphoblastoid lines? Is this a fair aggregation of data? Does the p-value listed refer to the difference in alleles? From the looks of the data, I wouldn't have expected a p value of 10^-13.

7) The conclusion that DNA polymorphisms in Treg-specific noncoding regulatory sites are enriched for association with autoimmune disease is not fully convincing. The association of these polymorphisms with total CD4^+^ T cell noncoding regulatory sites is far more impressive. Treg cells are a subset of CD4^+^ cells, and a “devil's advocate” would raise the question of whether Tregs might even be less likely than other types of CD4^+^ T cells to have their specific regulatory elements contribute to disease. The association is stated without discussing real alternatives in this version of the manuscript. Is the right null hypothesis being tested in Figure 3?

8) The figures do not explicitly show much evidence for a disease association among the Treg-specific enhancers. There is differential H3K27 acetylation, but the Venn diagrams shown seem to indicate that only a few Treg-specific elements are found among the many disease-associated SNPs. Is the Treg-specific element containing a SNP at the ENTPD1 locus disease-associated?

9) It seems surprising to use enhancer activity marks in the B lymphoblastoid cell lines as an additional source of statistical power for Treg-specific enhancers, e.g. in Figure 3 and elsewhere as described in Methods (under “Allele-specific chromatin modifications”). What does this statement mean? These marks should generally be cell type specific.

10) Legend to Figure 3—figure supplement 1: For interested biologists in the readership, explain what a Q–Q plot is and what the axes represent. It is a supplementary figure and the legend is the only source of information about what it is showing. Legends to panels F, G, H, and I are not legends but statements of interpretations of the data shown without explaining the plots themselves.

---

## [Author Response]

Essential revisions:

*1) A major motivation of this work is based on the statement, ‘it remains unknown if changes in the chromatin state of regulatory elements associated with cell lineage specification are conserved in their lineage-specificity’. However, this precise question was the major focus of Stergachis et al. Nature 515, 356-70, 2014. The Stergachis et al. paper, which is not cited, systematically analyzed the cis-acting circuitry of a large number of mouse and human cell types. A main point of Stergachis et al., is that while the precise genomic sequences containing cis regulatory elements for key transcription factors may not be conserved, the higher level organization of transcriptional networks is highly conserved. Although not the main focus of the paper, Treg cells are included in this analysis in*
Figures 3, 4 and 5*. The methods differ in that Stergachis et al. primarily rely on DNase hypersensitivity, which indicates TF binding, whereas the current manuscript uses H3K27ac to identify regions of regulatory activity. The current findings may be fully consistent with those of Stergachis, but at this point it is not clear the conclusion in the Abstract that ‘our findings suggest that a small set of regulatory elements specify the Treg lineage’ is correct. This conclusion is drawn from the data shown in*
Figure 2
*and*
Figure 2—figure supplement 1*, and is based on the relatively limited set of ‘conserved’ elements. These studies do not exclude the possibility that there are many non-conserved elements (that may represent conserved functional modules in different genomic locations) that are important for expression of Treg-specific genes. The authors will need to directly compare and contrast their findings and conclusions with those of Stergachis et al. with respect to the point of whether a small set of regulatory elements is sufficient to specify the Treg lineage.*

Our results are not in disagreement with the study by Stergachis et al. (Nature, 2014). We focus exclusively on the small number of lineage-specific regulatory elements and the conservation of their lineage specificity. In contrast, Stergachis et al. present conclusions regarding genome-wide transcriptional control programs of hundreds of transcription factors inferred from DNase footprints. Since we focus on the small number of lineage-specific elements, we are not able to perform the type of analysis reported by Stergachis et al., which requires aggregation across thousands of regulatory elements to achieve sufficient statistical power and biological robustness.

Nevertheless, to address the question of “can individual cell lineage-specific regulatory elements be non-conserved while enabling a similar conserved regulatory network”, we were able to examine the rate at which lineage-specific regulatory elements are lost and/or re-appear as independent regulatory elements near the same gene. This analysis involved three questions regarding “movement” of lineage-specific epigenetic elements within a genic locus, which we defined as the closest gene TSS and genes within 100kbp: (1) Is there mobility of lineage-specific epigenetic elements within a locus? (2) Do lineage-specific epigenetic elements become epigenetically active, but lineage non-specific, conserved genetic elements? (3) Would loosening of cutoffs for lineage-specificity and taking the maximally lineage-specific element across a large locus reveal conservation of the lineage specification program across non-conserved regulatory elements?

For (1), we identified genic loci that contained movement of lineage-specific epigenetic elements in mouse and human (Figure 2—figure supplement 2, example shown in Figure 2—figure supplement 2). Gene expression was concordant for 3/45 elements examined (not statistically significant).

For (2), a handful of lineage-specific elements became epigenetically inactive in the other organism (<15, not statistically significant) and had limited lineage-specific gene expression (Figure 2—figure supplement 2).

For (3), loosened rank cutoffs identified weakly lineage-specific elements in human (n = 632/496, up/down) and mouse (n = 759/625, up/down) at proximal but not necessarily orthogonal gene loci. Nearby genes were largely unaffected or exhibited discordant lineage-specific gene expression, indicating that this set may have limited functional relevance. Nearby genes with weak levels of differential expression (>0.5-fold or q < 0.01, where q is FDR value) are shown in Figure 2—figure supplement 2 (n = 26 up; n = 22 down), which includes many genes with regulatory elements that are lineage specific in both mouse and human as characterized in Figure 2—figure supplement 2.

We now include these analyses as a new supplemental figure (Figure 2—figure supplement 2) and describe these results in the text. We also discuss implications regarding Stergachis et al. observation of broad conservation of non-lineage-specific regulatory networks.

*2) Many of the figure panels in this manuscript are hard to interpret, and are inadequately explained in the legends. Specific examples of this are detailed in ‘Additional concerns’. Secondly, the Methods section does not give enough detail about the analyses that were performed to explain how certain numbers were arrived at. There needs to be more detail about what computational processes and parameters were used. Ideally, code would be provided as well to enable thorough review of what is largely a data-analysis, rather than data-generation, paper.* We have extended figure legends, descriptions of methods, and provided code.

Specifically, we now provide:

a) Main text heatmap quantifying nearby gene expression of the most lineage-conserved acetylated loci. This could previously be derived from supplemental tables, but is now made immediately accessible to reader. Additional heatmaps in supplemental Figure 2—figure supplement 2 expand on this analysis and supplemental tables contain raw data;

b) Code and libraries. The broad code layout is described via top-of-file and inline comments. The code is written in the portable R language. We have provided both original and clean linear code for most of the main analyses, since actual code behind this study (also provided) is > 10K lines with ∼80% being exploratory. We also provide ∼10K lines of code from custom libraries. All code and cached data have been deposited to a BitBucket git code repository;

c) We have reorganized figures and greatly expanded details in legends. We added text describing statistical methods, code, and data processing to ensure reproducibility of results;

d) We have added text discussing the relationship of our lineage-specific analysis results with the existing literature describing global epigenetic and/or genetic conservation;

e) We have expanded the Methods section to include additional details and analyses.

*Thirdly, many of the claims relied on single examples, and many figures lacked sufficient numerical or statistical underpinnings. Also, there should be more discussion about the relationship between this analysis and the results in*
[27]*,*
[19]
*and*
[62]*.*

We thank the reviewers for requesting these details and agree that including more details improves the manuscript. We have expanded the description of our statistical methods in the main text, legends, and supplemental material. We note that all single examples have corresponding genome-wide analysis (e.g., Figure 5 has Figure 5 as genome-wide analysis). An example where a single example is employed is in the context of linking Treg cells with disease through epigenetic overlap with GWA study significant SNP associations. The single example proof of concept at CTLA4 (now Figure 5) demonstrates that a cluster of disease-associated polymorphisms are found in Treg lineage-specific enhancers, which acts as both an example and as proof of concept that if the polymorphisms in this cluster are functionally altering disease course, than they most likely do so through differential Treg cell function. This argument is expanded in our response to points 7 and 8.

*3)*
Figure 2
*show that only a few genes show correlation between K27 levels in human and mouse, and only a few genes show correlation in expression between human and mouse. It would seem that a listing of the genes in common between these two analyses would make the analysis much more meaningful.*

We thank the reviewers for requesting this table and agree that it provides the reader with concrete examples summarized in other figures. We have selected the set of genes showing consistency across organisms (Figure 2). For future meta-analyses, we provided the complete dataset with all regulatory elements and genes in both organisms in Supplementary files 5 and 11.

*4)*
Figure 2
*is a very strong demonstration that sites bound by Foxp3 only in human Treg cells lost H3K27ac only in humans, and conversely sites bound by Foxp3 only in mice lost H3K27 only in mice. In conserved sites both human and mice lost H3K27ac. This appears to support the notion that Foxp3 is associated with repression, but there is no comment on this. Would the authors discuss?*

We thank the reviewers for requesting further discussion, as we are very interested in the Foxp3-mediated transcriptional control program, critical for Treg cell differentiation and function. We have added text that frames this observation in the context of our previous work, which suggested as the reviewers noted that Foxp3 may act predominantly as a negative regulator of transcription (Arvey et al., Nature Immunology, 2014).

*5)*
Figure 3
*expounds on the ENTPD1 gene encoding the ectonucleoside triphosphate diphosphohydrolase, CD39. They show and state, “ENTPD1 locus demonstrated genetic association with H3K27ac levels across individuals and in an allele-specific manner in a given individual (*Figure 3*).” The odd thing about this is that CD39 is ubiquitously expressed in virtually all tissues. Although the authors show an association with H3K27 levels, they did not show whether this translated into changes in gene expression. I would think the authors should address this issue, especially as CD39 is so widely expressed.*

*9) It seems surprising to use enhancer activity marks in the B lymphoblastoid cell lines as an additional source of statistical power for Treg-specific enhancers, e.g. in*
Figure 3
*and elsewhere as described in Methods (under “Allele-specific chromatin modifications”). What does this statement mean? These marks should generally be cell type specific.*

The reviewers raise multiple good points. We have added text to better explain our reasoning and claims.

With respect to *ENTPD1* lineage specificity, we acknowledged in original manuscript that CD39 has diverse expression, including lymphoblastoid cell lines that are used to confirm our observation in Treg cells. However, it is worth noting the previous immunophenotyping studies have identified genetic association that influences Treg-specific expression of ENTPD1 protein (41).

With respect to *ENTPD1* gene expression, we note that *ENTPD1* is associated with a cis-eQTL in Treg cells (Ferrera et al., 2014, PNAS). To our knowledge, we are the first to note specific regulatory elements with H3K27ac SNP QTLs, which we anticipate will be important for narrowing the search for causative polymorphisms.

Furthermore, *ENTPD1* has been implicated in Treg cell suppressor function by facilitating conversion of extracellular ATP, a pro-inflammatory mediator, into adenosine, a potent anti-inflammatory mediator acting upon A2A receptors expressed on different immune cell types including T cells, myeloid cells, and dendritic cells. It is important to emphasize that all known and putative mediators of Treg cell function are expressed in different cell types, yet have non-redundant function in Treg cells. We have now included into the manuscript this additional reasoning behind emphasis on *ENTPD1*.

*6)*
Figure 3
*shows the K27ac reads associated with two alleles in the same individual, which is a nice analysis; however, do the points represent individual people? How many are people, and how many are lymphoblastoid lines? Is this a fair aggregation of data? Does the* p*-value listed refer to the difference in alleles? From the looks of the data, I wouldn't have expected a* p *value of 10^-13.*

The reviewers raise an excellent question, which offered us an opportunity to clarify and expand on the goals, details, and results of our study.

The majority of points in Figure 3 represent lymphoblastoid cell lines (18 out of 23). The p-value referred to an aggregated binomial test of allele-specific heterozygous individuals and t-test of homozygous individuals. A regression model summarizing both homozygous and allele-specific heterozygous H3K27ac gives consistent results (see Methods for details).

While we believe this is not an unfair aggregation of data, it is by no means an ideal aggregation of data. However, our study is woefully underpowered to identify allele-specific chromatin modifications. We thus identified nominally significant loci in Treg cells and then used LCLs as a confirmation dataset.

These points have now been added to the main text and figure legends.

*7) The conclusion that DNA polymorphisms in Treg-specific noncoding regulatory sites are enriched for association with autoimmune disease is not fully convincing. The association of these polymorphisms with total CD4*^*+*^
*T cell noncoding regulatory sites is far more impressive. Treg cells are a subset of CD4*^*+*^
*cells, and a “devil's advocate” would raise the question of whether Tregs might even be less likely than other types of CD4*^*+*^
*T cells to have their specific regulatory elements contribute to disease. The association is stated without discussing real alternatives in this version of the manuscript. Is the right null hypothesis being tested in*
Figure 3*?*

We absolutely agree with the reviewers that CD4^+^ T effector cell function may be an equal or larger determinant of autoimmunity than Treg-specific dysfunction. We have included this sentiment in Discussion, including that for most autoimmune diseases, MHC class II HLA haplotypes are typically by far the most significant association than any other loci in the genome. The altered antigen presentation most likely has broad effect on CD4^+^ T cell compartment, including effector populations, as has been shown for example in type 1 diabetes (Marrack and Kappler 2012, [51]).

We present evidence that Treg cell dysfunction may contribute to pathogenesis in addition to the contribution of other lymphocyte populations. For instance, the *CTLA4* locus, which is assayed using the high resolution ImmunoChip and highlighted in Figure 5, presents evidence that Treg cells are potentially involved in T1D pathogenesis. Other genetic evidence links Treg cells to disease (e.g., at *IL2RA*) but lacks genetic resolution due to linkage and/or Treg-exclusive regulatory elements. However, *IL2RA* does have Treg upregulated epigenetic elements and these are in LD with disease-associated polymorphisms that have been shown to have functional impact on regulatory T cell phenotype (see Garg et al., 2012).

8) The figures do not explicitly show much evidence for a disease association among the Treg-specific enhancers. There is differential H3K27 acetylation, but the Venn diagrams shown seem to indicate that only a few Treg-specific elements are found among the many disease-associated SNPs. Is the Treg-specific element containing a SNP at the ENTPD1 locus disease-associated?

*ENTPD1* is associated with several immunological phenotypes, including response to HIV and inflammatory disease (Nikolova et al. 2011, Friedman et al., 2009). ENTPD1 expression is also associated with enhanced Treg cell suppressive function (Rissiek et al., 2015); however, it is unclear if the genetic associations influence ENTPD1 in a Treg cell-intrinsic fashion to contribute to disease pathogenesis or through alternative cell lineages. Additionally, the epigenetic and genetic structure at the locus makes it challenging to characterize cell-intrinsic contributions to disease (also discussed in response to points 5 and 9 above). In contrast, *CTLA4* has genetic and epigenetic structure that makes it more tractable to assign true lineage specificity across the disease-associated haplotype block overlapping Treg cell-specific regulatory elements.

9) We have grouped points 5 and 9. Please see our response above at point 5.

*10. Legend to*
Figure 3—figure supplement 1*: For interested biologists in the readership, explain what a Q–Q plot is and what the axes represent. It is a supplementary figure and the legend is the only source of information about what it is showing. Legends to panels F, G, H, and I are not legends but statements of interpretations of the data shown without explaining the plots themselves.*

Legends have been dramatically expanded throughout the manuscript and supplemental material to include details of analysis in addition to interpretations.

References:

Friedman DJ, Künzli BM, A-Rahim YI, et al. From the Cover: CD39 deletion exacerbates experimental murine colitis and human polymorphisms increase susceptibility to inflammatory bowel disease. Proc Natl Acad Sci U S A. 2009;106(39):16,788–16,793.

Garg G, Tyler JR, Yang JH, et al. Type 1 diabetes-associated IL2RA variation lowers IL-2 signaling and contributes to diminished CD4+CD25 + regulatory T cell function. J Immunol. 2012;188(9):4644–4653.

Marrack P, Kappler JW. Do MHCII-presented neoantigens drive type 1 diabetes and other autoimmune diseases? Cold Spring Harb Perspect Med. 2012;2(9):a007765.

Nikolova M, Carriere M, Jenabian MA, et al. CD39/adenosine pathway is involved in AIDS progression. PLoS Pathog. 2011;7(7):e1002110.

Qu HQ, Bradfield JP, Grant SF, Hakonarson H, Polychronakos C, Consortium TIDG. Remapping the type I diabetes association of the CTLA4 locus. Genes Immun. 2009;10 Suppl 1:S27-32.

Rissiek A, Baumann I, Cuapio A, et al. The expression of CD39 on regulatory T cells is genetically driven and further upregulated at sites of inflammation. J Autoimmun. 015;58:12-20.